# Slag Substitution as a Cementing Material in Concrete: Mechanical, Physical and Environmental Properties

**DOI:** 10.3390/ma12182845

**Published:** 2019-09-04

**Authors:** María Eugenia Parron-Rubio, Francisca Perez-Garcia, Antonio Gonzalez-Herrera, Miguel José Oliveira, Maria Dolores Rubio-Cintas

**Affiliations:** 1Departamento de Ingeniería Industrial y Civil, Universidad de Cádiz, 11205 Algeciras, Spain; 2Departamento de Ingeniería Civil, Materiales y Fabricación, Universidad de Málaga, 29071 Málaga, Spain; 3Civil Engineering Department, University of Algarve, 8005-139 Faro, Portugal

**Keywords:** concrete, slag, valorisation, cement, circular economy

## Abstract

A circular economy is a current tenet that must be implemented in the field of construction. That would imply the study of the possibilities of the use of waste generated, for obtaining materials the used in construction as replacements for the raw material used. One of these possibilities is the substitution of the cement by slag, which contributes to the reduction of cement consumption, decreasing CO_2_ emissions, while solving a waste management problem. In the present paper, different types of concrete made by cement substitution with different type of slags have been studied in order to evaluate the properties of these materials. Cement is replaced by slag from different steel mills, both blast furnace and ladle furnace slag. The percentages of slag substitution by cement are 30%, 40% and 50% by weight. Mechanical, physical and environmental properties have been evaluated. Compressive and flexural strength have been analysed as the main mechanical properties. As far as physical properties go, density and porosity tests were be reported and analysed, and from an environmental point of view, a leachate study was performed. It has been found that some kinds of slag (blast furnace slag) are very suitable as substitutes for cement, providing properties above those of the reference concrete, while other types (ladle furnace slag) could be valid for non-structural applications, contributing in both cases to a circular economy.

## 1. Introduction

A circular economy is a currently accepted tenet, in which the traditional linear economy is transformed into a circular economy, where every activity is conceived as a cycle, where waste materials are considered as potential new resources, instead of by-products to discard.

In the field of construction, the challenge is to exploit the possibilities of the waste generated in the building industry as raw materials to be integrated in the same construction cycle. 

One of the fields where this strategy is feasible is the incorporation of the slag generated during the steel production into concrete production. It has been used in many processes in the cement production and paving industries. It is interesting to focus attention on this problem and to study, thoroughly, all the possibilities that the steel by-product presents. 

The substitution of the cement by slag provides two clear advantages; the first one is the use of a waste that must be managed in a landfill, and the second one, even more relevant, is the reduction in cement consumption, so the reduction of CO_2_ emissions needed for its production. 

Nowadays, there is already a lot of research that supports the adequacy of steel slag for the production of cementitious matrices [1,2,3,4,5,6]. Additionally, many studies in which aggregates are replaced by those types of by-products exist; e.g., blast furnace slag, copper slag, electric arc and fume dust have been used [2,7,8,9,10,11,12,13,14,15,16].

There are also some studies on the substitutions of cement by ground granulated blast furnace slag (GGBFS) [5,17,18,19,20], even getting up to 80% of the cement removed by this type of slag. Khatib et al. [21] replaced up to 80% of cement by GGBFS making different substitutions. Good results were obtained in the substitutions up to 60%, since compressive strengths similar to conventional concrete were obtained. After 28 and 90 days, the strength was increased. Nevertheless, worse results were obtained when replacing 80% of the slag, and in the first days of setting, the strength of the reference concrete was not reached.

Less attention has been paid to the substitution of cement by ladle furnace slag (LFS). In previous works, different types of slag have been studied and compared, with a maximum amount of 25% of cement replaced [22]. These studies provided promising results. 

In this paper, different types of concrete have been elaborated on, in which the cement is replaced by slag from different steel mills, both blast furnace and ladle furnace slag.

The percentages of slag substitution by cement were 30%, 40% and 50% by weight. The substitution of cement was made in each mix by types of slag from different factories in Spain. According to different studies, it is known that the component with the highest influence over the durability of cementitious mixtures is SiO_2_. In this work, we will focus on the relationship between the amount of this component in each slag and the mechanical properties. 

Compressive and flexural strength were analysed as the main mechanical properties, making a comparison between all of them to evaluate which one provides the best characteristics. 

Additionally, some of the physical and environmental properties evaluated were included in the present paper. For physical properties, density and porosity test were reported and analysed. For a sake of brevity, other tests made are omitted. From an environmental point of view, a leachate study of the material was carried out, since it was essential, considering that waste material was being put into service. 

The paper is structured as follows. In Section 2 and Section 3, the materials studied and the tests performed have been briefly described. A longer Section 4 is devoted to the results obtained, along with a broad discussion with a special attention to the analysis of the mechanical properties. Finally, conclusions are outlined in Section 5.

## 2. Materials

In this work, a 52.5 R Portland cement (PC) is used; this cement was chosen as it is free of any additives; that is to say, composed of clinker between 95%–100% and between 0%–5% of minor components, without other additives that change their composition. The substitutions were made for the different types of slag that are shown below:➢Slag 1 (GGBFS): Granulated blast furnace slag ground in ball mill.➢Slag 2 (LFS1): Ladle furnace slag (LFS).➢Slag 3 (LFS2): Ladle furnace slag (LFS).

LFS1 and LFS2 are ladle furnace slag with different origins and composition. Slag 1 (GGBFS) has a particle size <0.063 μm provided by the company, while the LFSs were sieved in the laboratory to obtain equal granulometry from them. This is an important fact to keep in mind, since it would be interesting to see what would happen if the LFS slags were also treated in the same way as the GGBFS, but the companies that provided us with that type of slag did not have the technology to do so. Therefore, it was decided to carry out the study with screened slags to study its results.

Three different concrete mixes were designed by substituting 30%, 40% and 50% of the weight of the cement with slag obtained from three different steel mills in Spain.

The most characteristic chemical values of these slags are shown in Table 1. These values were determined by X-ray fluorescence (XRF). This test was performed with the LFS slags once screened.

Table 1 shows the major components of the slags studied; the rest of secondary compounds are described in another paper which used the same slags [22].

The main components (CaO, SiO_2_, Al_2_O_3_) of each of the slags are transcribed on a ternary diagram (Figure 1). Observe how the blast furnace slags are those that have better pozzolanic properties, by containing a higher percentage of SiO_2_.

The different concrete mixtures were named as follows:Mix 1 (MPC): Ordinary concrete without slag.Mix 2 (MGGBFS): Concrete with 30%, 40% or 50% cement replaced with processed slag.Mix 3 (MLFS1): Concrete with 30%, 40% or 50% cement replaced with unprocessed slag.Mix 4 (MLFS2): Concrete with 30%, 40% or 50% cement replaced with stainless steel slag.

Table 2 shows the dosages and the percentages of substitution to be made in each mixture. The W/C (Cement water ratio) (ratio is 0.5; the tests are the continuation of the paper belonging to this research group, in which only substitutions were made up to 25% [22]. Additional details can be found in that reference.

## 3. Tests’ Descriptions

Concrete mixes defined in the preceding section were subject to different tests. The main objective of these tests was to evaluate the effects on the mechanical characteristics (flexural and compressive strength), when cement is replaced by slag.

Concrete was made with the proportions shown in Table 2, where a 30%, 40% and 50% of the PC was substituted by the different slag according to Table 1, providing the different samples previously described.

The different mixture proportion was made according to the EN 12390-2 norm [23] for testing hardened concrete.

### 3.1. Physical Properties

The densities and porosities of the new materials were studied. They were obtained according to the EN 12390-7 norm [24].

A cubic specimen of 10 cm of edge were used for this test. Two specimens for each type of concrete were tested. The determination of the parameters was made for concrete of more than 28 days of age.

The formulation that was used to obtain the parameters is the following:(1)$$\mathrm{Density}\;\;D=\frac{P_{s}}{P_{\mathrm{sss}}-P_{\mathrm{sum}}}$$

(2)$$\mathrm{Porosity}\;P=100\frac{P_{\mathrm{sss}}-P_{s}}{P_{\mathrm{sss}}-P_{\mathrm{sum}}}$$

The parameters are obtained in the following way:Psum: Weight obtained by the hydrostatic balance (submerged weight), placing the specimen inside. This test piece must be completely saturated with waterPsss (saturated surface dry weight): Obtained by drying the surface water with a damp cloth.Ps (dry weight): It is obtained by drying the test pieces in the oven and checking every 24 hours that the mass loss is not less than 0.2%, at a temperature of 105 ± 5 °C. As indicated in the EN_12390-7 [24] standard, to carry out the test, the hydrostatic balance is used, to which a basket is attached where the test piece is introduced. In that way, one obtains the weight of the submerged sample.

### 3.2. Mechanical Properties

In order to obtain the compressive strength of the concrete specimens, we used cubes with edges of 10 cm and an automated press with a 2.000 kN capacity. The specimens were made according to the normative EN 12390-3 [25] and EN 12390-4 [26]; the fresh mixes were vibrated on a vibrating table and they were cured in a water bath 20 ± 2 °C. Then, they were tested at the ages of 1, 7, 28 and 90 days. For each of the mix proportions (Table 2), three different mixtures were made, and two specimens were tested at the different concrete ages (see reference [16] for details).

For the flexural strength, tests prismatic specimens were used with dimensions 4 × 4 × 16 cm^3^, made of the same kneaded as for the rest of the trial. That parameter was calculated using the uniform application of centred load. The same type of curing as the compressive test specimens was applied. This test was performed at 28 and 90 days. The prismatic test specimen was subjected to a bending moment by applying a load through upper and lower rollers, registering the maximum applied load calculating the flexural strength by EN-12390-5 [27] 

### 3.3. Leachate

Finally, the mixtures are subjected to leachate tests. The cement substitution percentage chosen was 30% for every mixture. Additionally, we carried the test out using a 50% slag substitution percentage for GGBFS, since throughout the research, it showed similar behaviour to conventional concrete. Specimens were immersed in 1 litre of distilled water for 2 days. The treatment that was made to the water sample in the laboratory was to sieve the sample with a 0.45 µm filter, and acidify them to pH < 2. Once that process was done, it was introduced into the spectrometer.

## 4. Results and Discussion

The results obtained for the physical (density and porosity), mechanical (compressive and flexural) and environmental (leachate) tests for each of the mixtures and their different substitutions are shown below, making a comparison between the percentages of loss and gains of strength, and the differences between the varieties.

### 4.1. Density

In the differently manufactured mixtures, it was observed (Table 3) that the density varied very little with respect to the standard mixture, decreasing only in the mixtures made with the slag LFS1; therefore, this indicates that LFS1 aerates the mixture more. It was significant, and the porosity of the material also increased significantly. For the mixtures with the other two types of slags, there were no significant differences; therefore, replacing them in the mixtures would not pose any problem for this property.

### 4.2. Porosity

The results of the porosity test are shown in the Table 4 and plotted in Figure 2. 

This property is tightly linked to the durability of the concrete. Taking as reference the PC mixture, for the mixtures with GGBFS, we see that it is practically constant for all the substitutions, even decreasing its porosity by 25% in the 50% substitution, with respect to conventional concrete. Regardless of the percentages of cement substitution, it was observed that the mixtures with slag LFS1 had a high porosity, since its porosity increased significantly. We saw in point Section 4.1, that for the mixture LFS1, the porosity increased up to 60%, so it was confirmed that this type of slag increases the incorporation of air into the concrete. On the contrary, it is observed that those made with slag LFS2, obtained a lower porosity, reaching up to a 43% improvement of results with respect to conventional concrete; this indicates that the links between particles that occur within the mixture are greater with this type of slag, without increasing its density by the same percentage; therefore, they provide better mixing conditions at the time of commissioning.

In short, the GGBFS and LFS2 mixtures’ lower porosity means a better performance in the long run, as this makes it more difficult for external agents to lead to the deterioration of the material, which affects the steel frame, in the case of reinforced elements. The opposite occurs with the LFS1 mixtures; it has a higher porosity, which can lead to the material having a shorter shelf life thanks to external agents that can damage it.

### 4.3. Compressive Strength

Results of mixtures for slag substitution.

#### 4.3.1. Slag GGBF

Figure 3 shows the compressive strength over time and how the GGBF slag influences concrete properties, from 1 day to 90 days of testing. These data are mean values for six samples per test (for the different concrete age). Results show that the compressive strength in the first days of hardening was lower than in conventional mixtures, becoming equal after 7 days and even increasing after 30 days. This is coincident with our previous results [16] and in accordance with the results reported by Wang [28,29].

Figure 4 shows the percentages of loss or increase in strength at day one and after 90 days. It is easy to visualize how, at one day there was up to 50% less strength than the conventional mixture. Nevertheless, after 90 days, it acquired up to 10% more compressive strength in the 30% replacement.

#### 4.3.2. Slag LFS1

Figure 5 shows the mean values obtained for the samples made with slag LFS1. This average has been made with six test specimens for each of the four mixtures.

The results show that in the LFS1 slag there was a lesser strength for day one, but in this case the compressive strength decreased over time. The loss at 90 days with a 50% substitution of cement was of almost 70% of PC’s strength, as shown in Figure 6. The loss of strength is maintained at both day one and after 90 days.

#### 4.3.3. Slag LFS2

Figure 7 shows the mean values for the specimens made of slag LFS2. Again, six specimens per mixture and age were tested.

With the slag LFS2, it was observed, as in the mixtures with slag LFS1, that there was a decrease of the strength both on day one, and over time. At 90 days and with a 50% replacement of cement, the loss was 40% (Figure 8).

The results lead us to think that, for high percentage substitutions, specimens with ladle furnace slag (LFS) substitutions have a higher strength loss compared with conventional concrete than those with blast furnace slag (GGBFS) substitutions. Particularly, the mixtures with LFS1 substitutions showed the worst behaviour at compressive strength tests, obtaining up to a 70% of compressive strength loss, when the cement substitution percentage was 50%.

#### 4.3.4. Comparison between the Mixtures

In Figure 9, the comparison of compressive strength for 90 day is shown for the different mixtures. In order to complete the figure, results previously reported with a 25% cement replacement [16] have also been included in the Figure.

It is observed how GGBFS slag presents an increase in strength; it provides pozzolanic benefits to the mixture. On the contrary, what occurs in those made with LFS slags has already been observed by other researchers, like Manso [30,31]. In this figure, it can be seen that the proportion of loss of strength is not the same in the two cases being the substitution up to 30% of LFS2 admissible for concretes with minor strength needs.

We can conclude that the blast furnace slag (GGBFS) is a good substitute for cement in terms of compressive strength. On the other hand, there is such a loss of strength on the other two ladle furnace slags (in the best of cases, 23%), that rule out any possibility of using this concrete as structural material. Nevertheless, they could be acceptable, being able to withstand medium-environmental pressure for situations in which the strength needs are lesser.

Returning to the ternary diagram in Figure 1, it is clear that a higher amount of SiO_2_, means better pozzolanic characteristics in the mixture, and in this case, it was observed that the mixtures with slag LFS1 were those that obtained the worst compressive strength, a cause not only of the greater porosity, and therefore greater amount of voids that weaken the mixture, but the slag also contained less SiO_2_.

### 4.4. Flexure Strength Tests

A similar study has been made to evaluate the flexure strength. In this case, the results obtained for all the mixtures are shown in Figure 10 and Figure 11, breaking the test pieces at 28 and 90 days, respectively.

Following a similar trend to the previous section, the mixtures made with ladle furnace slag (LFS) obtained worse strength than the mixture without substitution. On the contrary, those made with blast furnace slag (GGBFS) were equal to and even improved the reference concrete.

This loss or increase in strength was observed not only at 28 days, but also at 90 days (Figure 10). It also shows that mixtures with substitutions LFS1 behave worse than LFS2; additionally, showing clear differences between them. This behaviour is due to the chemical composition of the slag.

The percentages of loss of flexure strength of each one of the cases are shown in Figure 12, where it is seen how the GGBFS contributes to the material, the same characteristics as the PC, increasing its strength by 4% with a substitution of 50%. 

### 4.5. Comparison between 90-Day Flexure and Compressive Strength

The significance of this study is highlighted in the comparison of the gain or loss of the strength against flexure and compression combined. We will focus on that point in this section.

In Figure 13, the comparison in percentage of the strength, both compressive and flexural, in each of the mixtures made, highlights that the losses of flexural strength of the mixtures with slag LFS are much lower than those of compressive strength. This decrease in strength is practically half in most cases. In the strength of the mixtures with GGBFS does not increase twice as much in the flexural strength, as would be expected by the previous results, but only half. Apart from substitutions of 50%, this percentage of 4% is maintained.

This indicates the feasibility of LFS slag in non-structural elements, valuing a waste as a by-product, reducing the production of cement that generates a large amount of CO_2_ into the environment.

In all the investigations where the laws of mechanical behaviour (constitutive models) for materials are established, outcomes are considered a representative volumetric element of the same. It is assumed that the material behaves as a continuous medium; that is, it has the same elastic properties at each point.

### 4.6. Leachate

The results obtained are shown in Table 5.

In general terms, it can be observed how most of the values obtained had a decrease in relation to conventional concrete. The most significant is to see how the amount of chromium in the mixtures with slag, compared with the conventional concrete mixture, decreased.

It is significant how one of the most harmful elements in the leachate is chromium, and this element decreases with respect to the master mix. This is interpreted as the encapsulation of the slag being diluted into the cementitious matrix absorbing this metal, without generating any environmental danger when it is used.

There are some values which are slightly above those obtained with the reference concrete (PC); however, all these leachate ranges fall within the values allowed by the Code of Federal Regulations (CFR) 40CFR/261.24. 

According to those maximum values the encapsulation of the slag in the concrete not only does not leach contaminant, but also reduces the leachate of one of the more dangerous components that are measured in the CFR—the Cr, not exceeding the limit of 5 mg/L. 

## 5. Conclusions

According to the results described, we can outline the following conclusions:→The results lead us to think that, for high percentage substitutions, specimens with ladle furnace slag (LFS) substitutions have a higher strength loss compared with conventional concrete than those with blast furnace slag (GGBFS) substitutions. Particularly, the mixture with LFS1 slag substitutions showed the worst behaviour in compressive strength tests, obtaining up to a 72% of compressive strength loss with a cement substitution percentage of 50%.→On the other hand, specimens with blast furnace slag substitution showed an increase in compressive strength of 10% at 90 days. Mixtures using this type of slag substitution showed a slower hardening process, with a compressive strength reduction at day one, but gaining a compressive strength similar to or even above the conventional concrete after 7 days.→The chemical characteristics of the slag influence the mixtures and strength. It was observed in this study, that for essential components such as SiO_2_, the lower the percentage, the lower the strength. As it can be seen in the mixes made with LFS1 and LFS2 slags, the lack of that compound makes them work worse. GGBFSs are the best performers, having twice the amount of that compound, increasing its strength even with respect to conventional concrete. The good pozzolanic activity that contributes to those types of cementitious mixtures was verified.→Another characteristic result of this investigation is the difference of compressive and flexural strength among the different mixtures. LFS presents a loss of flexural strength that is the half of the loss of compressive strength. This suggests that they could be used in other fields of civil engineering with lower strength requirements.→Leaching test confirmed that slag does not cause damage to the environment. The results of the leaching tests of concrete mixtures with slag are similar to the results of traditional concrete. Therefore, slag encapsulation into the concrete seems to be a good strategy to manage this waste product, instead of it being deposited in landfills where it will pollute the environment by leaching.

As a final conclusion, it is clear that blast furnace slag (GGBFS) is suitable for the production of a sustainable concrete, and as a substitute for cement, since it has been proven to bring the same characteristics to the mixture as cement. 

For the other two types of mixtures with slag (LFS), a non-structural application would be suitable. This would put a value on the residue, avoiding the consumption of raw material and reducing the landfill deposit.

Concrete production with slag is a clear example of circular economics, since the steel manufactured, necessary to build structures, generates waste that can be incorporated to the same building cycle.

## Figures and Tables

**Figure 1 materials-12-02845-f001:**
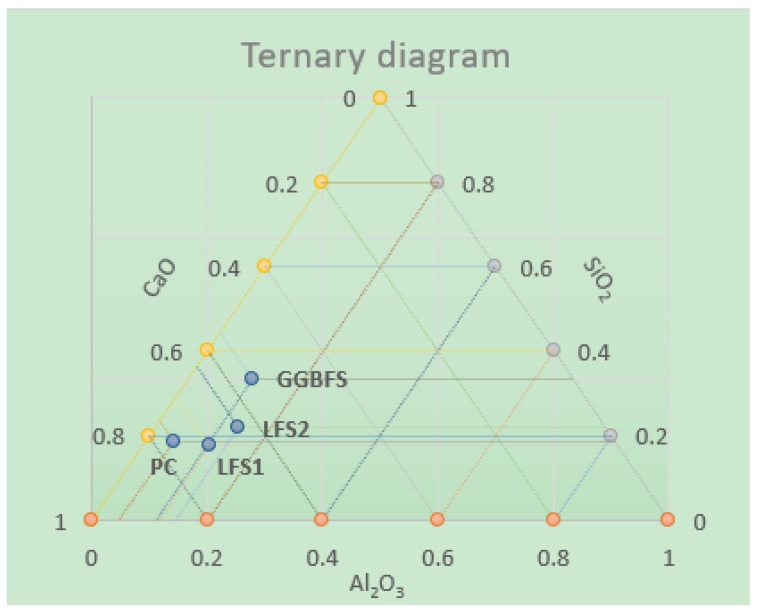
Ternary diagram indicating the compositions of Portland cement (PC), ground granulated blast furnace slag (GGBFS) and ladle furnace slag (LFS) in the system.

**Figure 2 materials-12-02845-f002:**
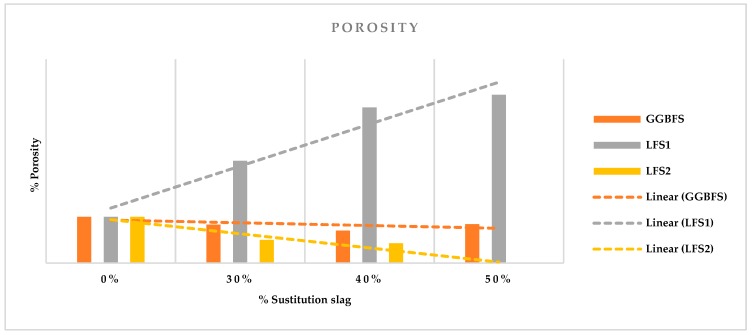
Porosity.

**Figure 3 materials-12-02845-f003:**
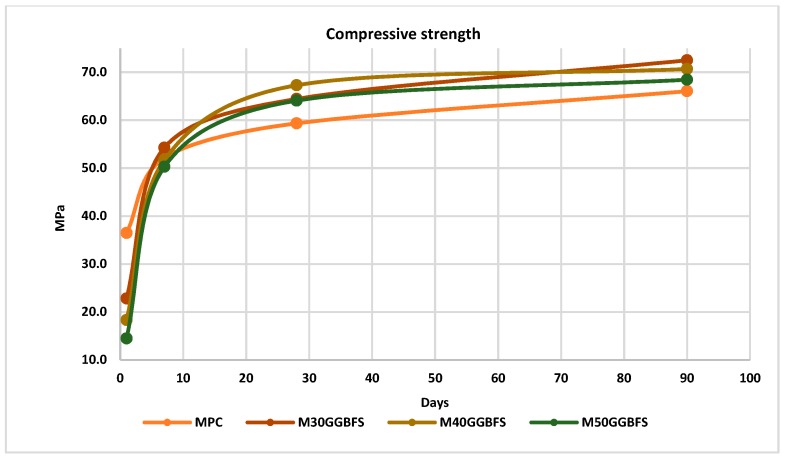
Compressive strength GGBFS.

**Figure 4 materials-12-02845-f004:**
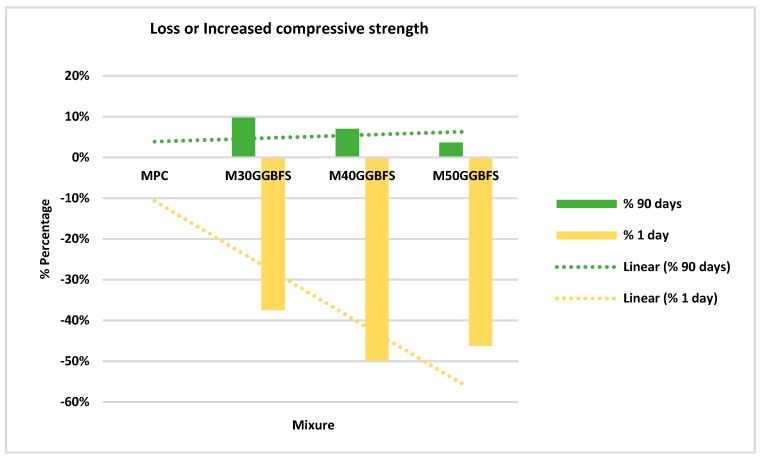
Percentage of loss or increase in compressive strength GGBFS.

**Figure 5 materials-12-02845-f005:**
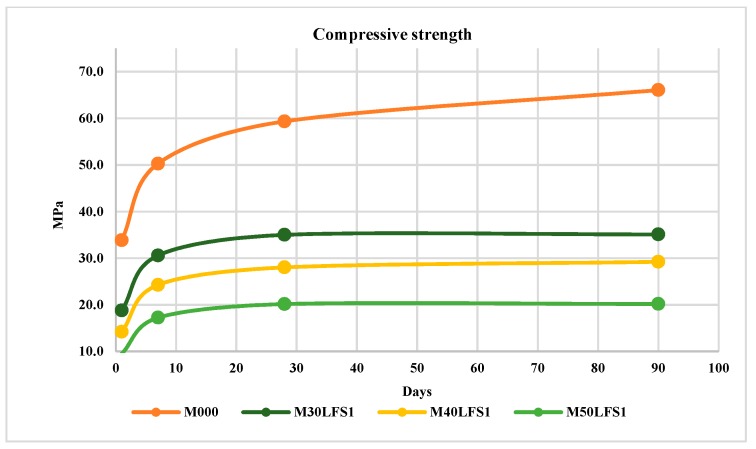
Compressive strength LFS1.

**Figure 6 materials-12-02845-f006:**
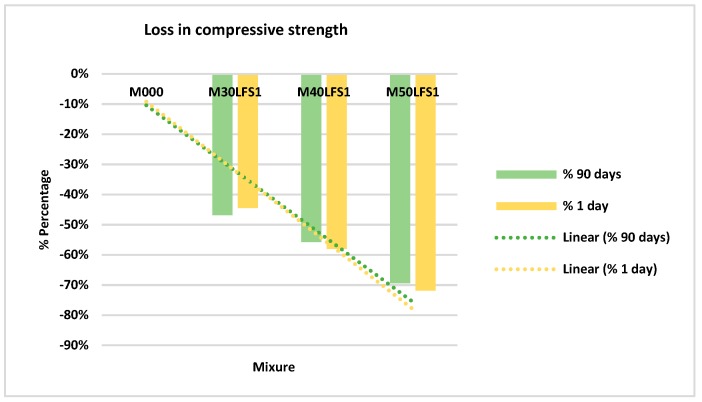
Percentage of loss in compressive strength LFS1.

**Figure 7 materials-12-02845-f007:**
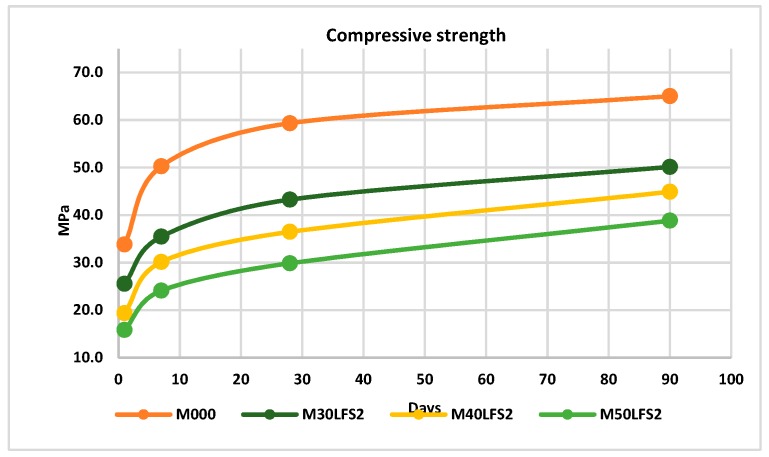
Compressive strength LFS2.

**Figure 8 materials-12-02845-f008:**
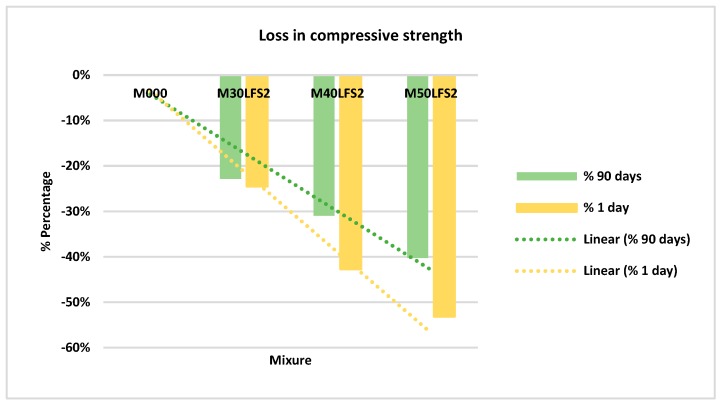
Percentage of loss in compressive strength LFS2.

**Figure 9 materials-12-02845-f009:**
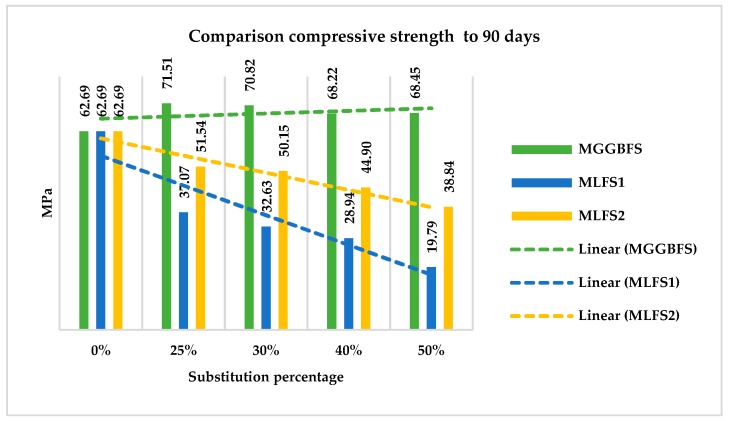
Comparison compressive strength to 90 days, for the three different slag.

**Figure 10 materials-12-02845-f010:**
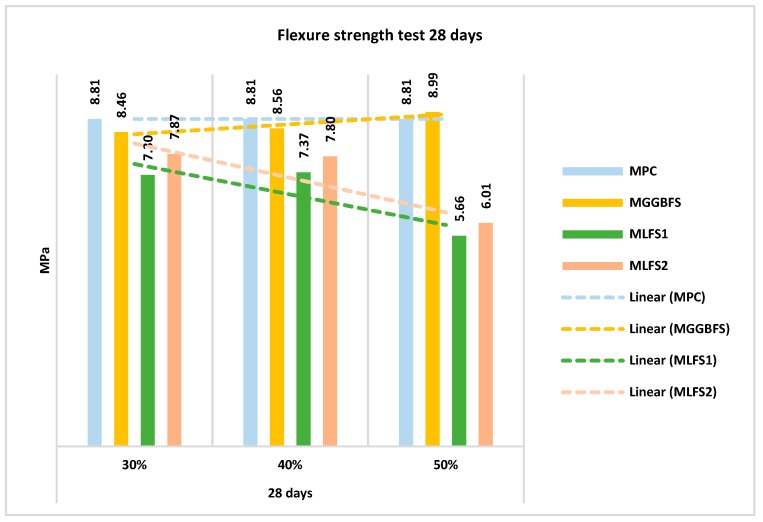
Flexure strength test 28 Days.

**Figure 11 materials-12-02845-f011:**
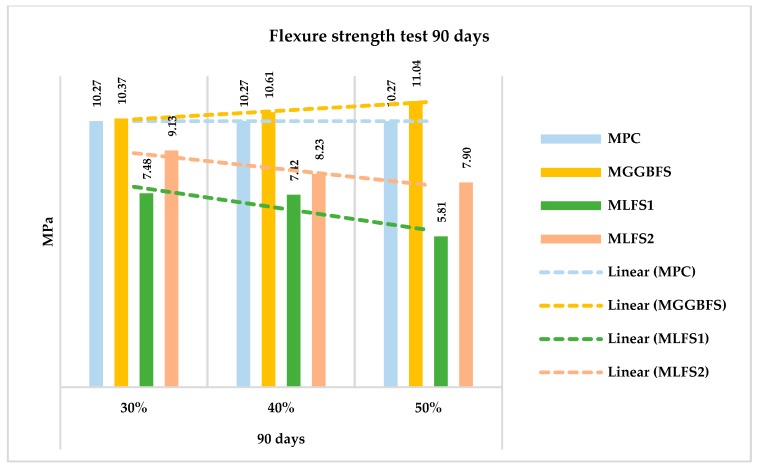
Flexure strength test at 90 days.

**Figure 12 materials-12-02845-f012:**
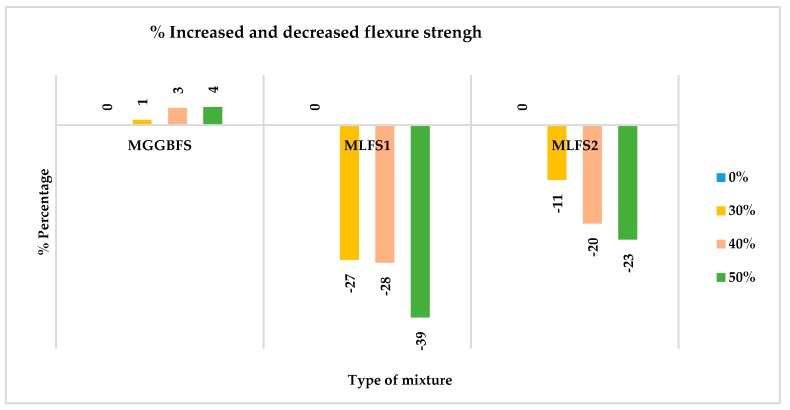
Percentage increased and decreased flexure strength.

**Figure 13 materials-12-02845-f013:**
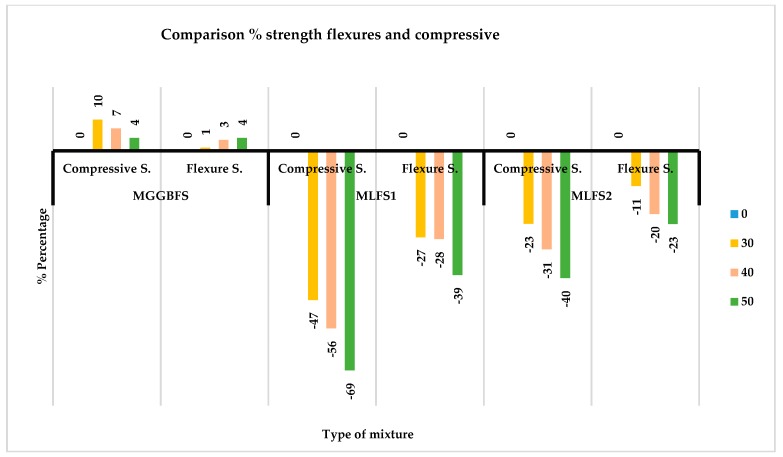
Comparison of flexural and compressive strength.

**Table 1 materials-12-02845-t001:** Cement and slag chemical composition.

Cement and Slag Origin/Chemical Composition	SiO_2_	Al_2_O_3_	Fe_2_O_3_	CaO	MgO
	%	%	%	%	%
PC	16.6 ± 0.5	4.25 ± 0.5	3.02 ± 0.02	67.92 ± 0.5	1.43 ± 0.05
GGBFS	32.3 ± 0.5	10.7 ± 0.5	0.29 ± 0.02	47.14 ± 0.5	7.64 ± 0.05
LFS1	13.7 ± 0.5	9.1 ± 0.5	1.57 ± 0.02	55.18 ± 0.5	16.9 ± 0.05
LFS2	18.8 ± 0.5	12.5 ± 0.5	2.34 ± 0.02	54.9 ± 0.5	6.99 ± 0.05

**Table 2 materials-12-02845-t002:** Concrete mixture proportion.

		Binder				Aggregates			
Mix	Water (*w*/*b* Ratio)	Dosage	Cement	Slag	Additive (Superpla-Sticizer)	Dosage	Fine Sand 0–2 mm	Sand 0–4 mm	Gravel 4–16 mm
MPC	0.5	300 kg/m^3^	100%	0%	3.9 kg/m^3^	2033.8 kg/m^3^	15%	35%	50%
M30GGBFS			70%	30%					
M40GGBFS			60%	40%					
M50GGBFS			50%	50%					
M30LFS1			70%	30%					
M40LFS1			60%	40%					
M50LFS1			50%	50%					
M30LFS2			70%	30%					
M40LFS2			60%	40%					
M50LFS2			50%	50%					

**Table 3 materials-12-02845-t003:** Density of the mixtures.

Density (kg/m^3^)				
	PC	GGBFS	LFS1	LFS2
30%	2500 ± 100	2490 ± 100	2450 ± 50	2530 ± 10
40%	2500 ± 100	2490 ± 100	2370 ± 30	2510 ± 10
50%	2500 ± 100	2490 ± 100	2370 ± 40	2510 ± 10

**Table 4 materials-12-02845-t004:** Porosity of the mixtures.

Porosity (%)				
	PC	GGBFS	LFS1	LFS2
0%	1.81 ± 0.5	-	-	-
30%	-	1.51 ± 0.2	4.01 ± 0.5	0.91 ± 0.1
40%	-	1.27 ± 0.2	6.1 ± 0.5	0.78 ± 0.1
50%	-	1.53 ± 0.2	6.60 ± 0.5	-

**Table 5 materials-12-02845-t005:** Leachate of the mixtures.

Chemical Element	PC	GGBFS (30%)	GGBFS (50%)	LFS1 (30%)	LFS2 (30%)	CFR 40/261.24 (mg/L)
**[Mg] µg/L**	20.2 ± 1.0	21.3 ± 0.2	28.1 ± 1.0	26.0 ± 1.2	31.3 ± 1.0	-
**[Si] µg/L**	8.60 ± 0.33	13.1 ± 1.0	13.3 ± 1	9.30 ± 0.40	7.90 ± 0.73	-
**[Ti] µg/L**	<0.100	<0.100	<0.100	<.0,100	<0.100	-
**[Cr_total_] µg/L**	15.6 ± 0.6	1.10 ± 0.05	0.394 ± 0.013	0.673 ± 0.040	0.162 ± 0.023	5
**[Mn] µg/L**	0.100 ± 0.010	0.180 ± 0.010	0.190 ± 0.04	0.150 ± 0.020	0.101 ± 0.01	-
**[Fe] µg/L**	18.2 ± 0.7	17.0 ± 0.2	10.6 ± 0.03	1.82 ± 0.03	8.23 ± 0.20	-
**[Ni] µg/L**	0.270 ± 0.030	0.510 ± 0.020	0.280 ± 0.012	0.150 ± 0.013	0.270 ± 0.021	-
**[Cu] µg/L**	1.92 ± 0.10	4.3 ± 0.36	8.80 ± 0.033	1.50 ± 0.03	4.00 ± 0.04	-
**[Zn] µg/L**	6.60 ± 0.20	6.10 ± 0.12	3.90 ± 0.20	2.92 ± 0.10	4.40 ± 0.14	-
**[As] µg/L**	<0.200	<0.200	<0.200	<0.200	<0.200	5
**[Cd] µg/L**	<0.100	<0.100	<0.100	<0.100	<0.100	1
**[Sn] µg/L**	<0.100	<0.100	<0.100	<0.100	<0.100	-
**[Pb] µg/L**	0.543 ± 0.022	2.02 ± 0.02	0.230 ± 0.010	0.190 ± 0.001	0.333 ± 0.002	-

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
