# Peer review of "Slag Substitution as a Cementing Material in Concrete: Mechanical, Physical and Environmental Properties"

_materials, 2019, doi:10.3390/ma12182845_

Round 1

Reviewer 1 Report

1. Slag replacement is a very common topic in achieving sustainability in the construction industry. The authors to have same motivation. What is the novel idea of slag replacement? Need to highlight it. 2. Introduction has too many paragraphs. Rewrite it. 3. In section 2, it’s mentioned… this cement is chosen as it is free of additions . What is free? This is not technically described. Use proper scientific language. 4. Table 1 can be put in Kg/m3 unit. 5. Porosity results were simply described. There is no proper explanation for it. 6. Similar comment for compressive strength. The results were not described properly. 7. The sustainability aspect can be descried with some recent papers. Use some articles • Additive manufacturing of geopolymer for sustainable built environment • Performance of high strength concrete made with copper slag as a fine aggregate • Rheological behavior of high volume fly ash mixtures containing micro silica for digital construction application • Effect of copper slag and cement by-pass dust addition on mechanical properties of concrete,

Author Response

Point 1: Slag replacement is a very common topic in achieving sustainability in the construction industry. The authors to have same motivation. What is the novel idea of slag replacement? Need to highlight it.

Response 1: Thank you for your question, fundamentally, the novel idea is the comparison of different types of slags from different furnaces (electric arc spoon and blast furnace) and see the differences that are generated in the concrete, physical and mechanical properties, manufactured with them as cement replacement.

Point 2: Introduction has too many paragraphs. Rewrite it.

Response 2: Paragraphs have been deleted and some of them have been rewritten.

Point 3: In section 2, it’s mentioned… this cement is chosen as it is free of additions . What is free? This is not technically described. Use proper scientific language.

Response 3: This issue has been developed specifying the characteristics of CEM I 52.5R cement.

 “In this work a 52.5 R Portland cement (PC) is used, this cement is chosen as it is free of additives, that is to say, composed of clinker between 95-100% and between 0-5% of minor components, without other additives that change their composition.”

Point 4: Table 1 can be put in Kg/m3 unit.

Response 4: This table is expressed in percentage; therefore it would not be possible to express it in Kg/m3.

Point 5: Porosity results were simply described. There is no proper explanation for it. 

Response 5: An additional paragraph has been added explaining what would happen depending on the porosity obtained in the different mixtures.

“In short, the GGBFS and LFS2 mixtures lower porosity means a better performance in the long run, as this makes more difficult for external agents to lead to the deterioration of the material thus affecting its steel frame in case of reinforced elements. The opposite occurs with the LFS1 mixtures, since it has a higher porosity, which can lead to the material having a shorter shelf life and external agents that can damage it.”

Point 6: Similar comment for compressive strength. The results were not described properly.

Response 6: In this case, a further discussion is made when the flexural strength results are shown. We have avoided to repeat here argument provided in the following section.

Point 7: The sustainability aspect can be descried with some recent papers. Use some articles • Additive manufacturing of geopolymer for sustainable built environment • Performance of high strength concrete made with copper slag as a fine aggregate • Rheological behavior of high volume fly ash mixtures containing micro silica for digital construction application • Effect of copper slag and cement by-pass dust addition on mechanical properties of concrete.

Response 7: Thank you very much for your contribution, we have taken it into account in the introduction of the paper.

Reviewer 2 Report

The article under review has little merit and the research presented in it does not contribute much to the knowledge about the use of slag in concrete technology, largely repeating the already known theses. The linguistic side of the publication is questionable. The text contains both errors and expressions indicating a lack of knowledge of the nomenclature used in concrete technology. One of the latter is the common use of the word "resistance" as a synonym of the word "strength" meaning the mechanical characteristic of concrete. The form of figures is also controversial. The adopted colours (e.g. light blue background and light grey letters) cause that the figures are very poorly legible on black and white printouts, and it should not be assumed that readers will use only the electronic version of the article. The diagrams consistently show the unit Mpa (instead of MPa), and the percentage values are described by two symbols % - one for the number on the axis and the other in the description of the axis. The latter symbol should be replaced by the value represented on the axis (e.g. 'change in compression strength') without specifying the unit.

The other weaknesses of the article are described below in approximately chronological order as they were discovered during the reading of the article.

1) In the summary, the authors write about the use of waste generated in the building process. Slag is not a material created in such a process.

2) The second sentence of the introduction is, in the opinion of the reviewer, largely untrue and should not serve as a part of the justification of the research.

3) The authors compare two types of slag (blast furnace slag and ladle furnace slag) with each other, but completely ignore the fundamental difference between them, which results from the fact that one of them is ground (which increases its reactivity) and the other only sieved. The question also arises what is the granulation of LFS and what part of it are particles too large which have been rejected during the screening procedure. The differences between the activity of the slags used are certainly influenced by their specific surface area, which, however, has not been given (and probably has not been studied either). The article also repeats the statement that the SiO2 content determines the pozzolanic activity of the slag, while apart from its content, its form (amorphous or crystalline) is of key importance.

4) It is not clear for what purpose Figure 1 was placed in the article, since what is depicted on it and what was mentioned in the article (differentiation of SiO2 content) can be read from Table 1. 

5) The composition of the slag used should be supplemented with secondary components as the LFS is the result of the refining of steel and contains impurities washed away from it which may affect the durability of concrete

6) The article does not contain anything about the aggregate used. Referring to an earlier article for this information is not the right approach.

7) The flexural strength tests were carried out using 4x4x16 cm prisms. It is not clear whether these specimens had the same aggregate composition (including coarse aggregate) or whether the tests were carried out on mortar specimens. In the latter case, the comparison of the two types of strength is questionable. Rather, the tensile strength test should have been carried out in the splitting test.

8) The presented test results do not contain any information about the standard deviations of the measured parameters. Without this information it is difficult to assess e.g. the situation in which an increase in replacement rate of cement with LSF1 from 30% to 40% results in a decrease in concrete density by 80 kg/m3, and a subsequent increase from 40% to 50% does not affect the concrete density at all.  

9) The concrete porosity test raises serious doubts. The specimens were dried far too short, so the porosity values obtained are unbelievably low. Drying should be carried out until a constant mass is obtained, as materials of different porosity dry at different rates. In addition, it is impossible to remove all unbound water within 24 hours at the specified temperature. The reviewer's experience shows that it usually takes at least 7-10 days to achieve this, and this time depends mainly on the tightness of the material.

10) The authors write about "poor pore distribution" whereas no pore distribution studies were conducted (e.g. mercury porosimetry). The only thing that was examined, and this was incorrectly done, was the total volume of the pores. The pore size distribution has not been determined, so there is no basis for concluding that porosity is adequate or not.

11. Table 5 is a transhipment of data. According to the reviewer, it is pointless to include information on the leachability of such elements as sodium, potassium or calcium. From the environmental point of view, only the information on the leaching of heavy metals or elements having a negative impact on the environment is important.

12) The second of the conclusions is obvious and trivial. In the fourth application, the conclusions on the potential use of the LFS are not justified by the observed differences between compression strength decrease/increase and flexural strenght.

Author Response

Point 1: The article under review has little merit and the research presented in it does not contribute much to the knowledge about the use of slag in concrete technology, largely repeating the already known theses. The linguistic side of the publication is questionable. The text contains both errors and expressions indicating a lack of knowledge of the nomenclature used in concrete technology. One of the latter is the common use of the word "resistance" as a synonym of the word "strength" meaning the mechanical characteristic of concrete. The form of figures is also controversial. The adopted colours (e.g. light blue background and light grey letters) cause that the figures are very poorly legible on black and white printouts, and it should not be assumed that readers will use only the electronic version of the article. The diagrams consistently show the unit Mpa (instead of MPa), and the percentage values are described by two symbols % - one for the number on the axis and the other in the description of the axis. The latter symbol should be replaced by the value represented on the axis (e.g. 'change in compression strength') without specifying the unit.

The other weaknesses of the article are described below in approximately chronological order as they were discovered during the reading of the article.

Response 1: Thank you very much for your comments to get a better paper, all of them have been considered and have been modified throughout the paper, such as putting MPa, changing the word strength etc ...

Point 2: In the summary, the authors write about the use of waste generated in the building process. Slag is not a material created in such a process.

Response 2: This point has already been changed so that it is better understood, thank you very much for your contribution.

“This imply the study of the possibilities of the use of waste generated, in obtaining materials used in construction as a replacement for the raw material used”

Point 3: The second sentence of the introduction is, in the opinion of the reviewer, largely untrue and should not serve as a part of the justification of the research.

Response 3: The introduction has been reviewed and some paragraphs have been deleted including that.

Point 4: The authors compare two types of slag (blast furnace slag and ladle furnace slag) with each other, but completely ignore the fundamental difference between them, which results from the fact that one of them is ground (which increases its reactivity) and the other only sieved. The question also arises what is the granulation of LFS and what part of it are particles too large which have been rejected during the screening procedure. The differences between the activity of the slags used are certainly influenced by their specific surface area, which, however, has not been given (and probably has not been studied either). The article also repeats the statement that the SiO2 content determines the pozzolanic activity of the slag, while apart from its content, its form (amorphous or crystalline) is of key importance.

Response 4: Thank you very much for your contribution, it is true that the way of obtaining each of the slags is not the same. In the paper a phrase has been introduced in which it is specified that they do not have the same treatment but that they still wanted to continue the investigation to see the result that is obtained, taking into account that if they had the same treatment these results may vary.

“This is an important fact to keep in mind since it would be interesting to see what would happen if the LFS slags were also treated in the same way as the GGBFS, but the companies that provide us with this type of slag do not have the technology to do so. Therefore, it has been decided to carry out the study with screened slags to study its results.”

Point 5: It is not clear for what purpose Figure 1 was placed in the article, since what is depicted on it and what was mentioned in the article (differentiation of SiO2content) can be read from Table 1. 

Response 5: The authors believe that the figure clarifies the results of the chemical composition of the different slags since the position of each of them is clearly seen in the ternary diagram according to the most significant chemical compounds in them.

Point 6: The composition of the slag used should be supplemented with secondary components as the LFS is the result of the refining of steel and contains impurities washed away from it which may affect the durability of concrete

Response 6: In order to avoid redundant information, already published in a previous paper, the authors have decided to present in the table the most abundant elements only. The reference is provided in case of interest.

“Table 1 shows the major components of the slags studied, the rest of secondary compounds are described in another paper which used the same slags [22]”

 Point 7: The article does not contain anything about the aggregate used. Referring to an earlier article for this information is not the right approach.

Response 7: The granulometry of each of the aggregates is in table 2, they are quarried without much interest for this investigation.

 Point 8: The flexural strength tests were carried out using 4x4x16 cm prisms. It is not clear whether these specimens had the same aggregate composition (including coarse aggregate) or whether the tests were carried out on mortar specimens. In the latter case, the comparison of the two types of strength is questionable. Rather, the tensile strength test should have been carried out in the splitting test.

Response 8: A phrase has been added in which it is specified that the specimens are the same kneaded.

“For the flexural strength, tests prismatic specimens were used with dimensions 4 × 4 × 16 cm3, made of the same kneaded as for the rest of the trial.”

 Point 9: The presented test results do not contain any information about the standard deviations of the measured parameters. Without this information it is difficult to assess e.g. the situation in which an increase in replacement rate of cement with LSF1 from 30% to 40% results in a decrease in concrete density by 80 kg/m3, and a subsequent increase from 40% to 50% does not affect the concrete density at all.  

Response 9: The tolerance values of the means taken at the time of density and porosity have been added to the tables.

Point 10: The concrete porosity test raises serious doubts. The specimens were dried far too short, so the porosity values obtained are unbelievably low. Drying should be carried out until a constant mass is obtained, as materials of different porosity dry at different rates. In addition, it is impossible to remove all unbound water within 24 hours at the specified temperature. The reviewer's experience shows that it usually takes at least 7-10 days to achieve this, and this time depends mainly on the tightness of the material.

Response 10:  Thank you for your input, there has been a transcription error.

“It is obtained by drying the test pieces in the oven and checking every 24 hours that the mass loss is not less than 0.2%, at a temperature of 105 ± 5 ° C. As indicated in the EN_12390-7”

Point 11: The authors write about "poor pore distribution" whereas no pore distribution studies were conducted (e.g. mercury porosimetry). The only thing that was examined, and this was incorrectly done, was the total volume of the pores. The pore size distribution has not been determined, so there is no basis for concluding that porosity is adequate or not.

Response 11: Thank you very much for your contribution, it is true that in this essay the pore distribution is not measured, that phrase has been changed, since it must have been a transcription error that the authors have not detected, therefore this sentence has been changed.

“has a high porosity, since its porosity increases significantly”

Point 12: Table 5 is a transhipment of data. According to the reviewer, it is pointless to include information on the leachability of such elements as sodium, potassium or calcium. From the environmental point of view, only the information on the leaching of heavy metals or elements having a negative impact on the environment is important.

Response 12: Little significance elements have been removed as suggested.

Point 13: The second of the conclusions is obvious and trivial. In the fourth application, the conclusions on the potential use of the LFS are not justified by the observed differences between compression strength decrease/increase and flexural strenght.

Response 13: We humbly consider that the second conclusion is relevant (an 10 % increase in compression strength when the cement is replaced by slag). The text has been revised to improve the clarity.

“On the other hand, specimens with blast furnace slag substitution showed an increase in compression strength of 10% at 90 days. Mixtures using this type of slag substitution showed a slower hardening process, with a compression strength reduction at 1 day but gaining a compression strength similar to or even above the conventional concrete after 7 days.”

Regarding the fourth conclusion, it is a mere observation about the strength loss shown by LFS mixtures. Even when it is significant, it is interesting to point out that the benefit of using a waste can justify the use of these mixture in cases of lower strength requirement. The text has also been rewritten to improve the clarity.

“Another characteristic result of this investigation is the difference on compression and flexural strength among the different mixtures. LFS presents a loss of flexural strength that is the half of the loss of compression strength. This suggest that they could be used in other fields of civil engineering with lower strength requirement.”

Reviewer 3 Report

The manuscript deals with the physical, mechanical and environmental properties of cement concrets made with Portland Cement replaced by different types of slags. The work is interesting since it concerns the possibile reuse of by-products in construction, with advantages for economy and environment. Some concerns the reviewer has about the preliminary characterisation based on porosity and desnity only, that seems not so much useful and quite bare. More important remarks: 

1.Please pay attention to the verbs (e.g. process that make=makes, the resistance were=was etc.), singular and plural (e.g. one of the field=fields etc.). A revision of language is required..

2. The blue (cyan)  background of figures makes them less readable.

3. Is it possible to give more information about slags? production, origin etc. Why the 2 LFSs are so different in behaviour? May the Authors specify exactly the slags used? They sometimes use the term EAF slag as equivalent to LF slag but they are not the same.

Minor remarks:

Page 2. electric arc=electric arc furnace (EAF) slag

Page 3. The different concrete mixtures were named as follow: How processed, unprocessed slags etc. are related with GGBFS, LFS etc.?

How was the substitution done? By weight? Have the Authors taken into account the different specific weight of the materials?

Section 3: flexure and compressive=flexural and compressive strength

two cubic specimen of 10 cm. I know that the specimens are small, but were they compacted or vibrated?

centered load. It would be better a 4 points bending test, since with 3PBT the specimen breaks into the middle, while with 4PBT it breaks in the central third, where the resistance is lower

3.3 chose=chosen

3.3 to filter the sample. Please check the sentence

4.2. better porosity. At this moment it is not clear if higher porosity means better performance so the term "better" cannot be used here. better=higher

 4.3 other two electric arc slag (LFS) . Please pay attention that EAF slag is something else.and usually gives very good properties to the mixtures

4.5 flexion=flexure

Table 5. Chromium is Cr6+? or total?

Table 5. Clarify what CFR is

Author Response

Response to Reviewer 3 Comments

The manuscript deals with the physical, mechanical and environmental properties of cement concrets made with Portland Cement replaced by different types of slags. The work is interesting since it concerns the possibile reuse of by-products in construction, with advantages for economy and environment. Some concerns the reviewer has about the preliminary characterisation based on porosity and desnity only, that seems not so much useful and quite bare. More important remarks: 

Point 1:  Please pay attention to the verbs (e.g. process that make=makes, the resistance were=was etc.), singular and plural (e.g. one of the field=fields etc.). A revision of language is required..

Response 1: Thank you very much for your contribution to the improvement of the paper, the errors detected in the text have been changed.El fondo azul (cian) de las figuras las hace menos legibles.

Point 2:  The blue (cyan)  background of figures makes them less readable.

Response 2: It is true that the graphics did not look good, the authors have changed the colors of the figures for better visibility.

Point 3:   Is it possible to give more information about slags? production, origin etc. Why the 2 LFSs are so different in behaviour? May the Authors specify exactly the slags used? They sometimes use the term EAF slag as equivalent to LF slag but they are not the same.

Response 3: Unfortunately the slag supply companies work with confidentiality and no more slag data can be given.

The faults detected in the slag denomination have been modified.

Point 4:   Page 2. electric arc=electric arc furnace (EAF) slag

Response 4: The observation has been modified

Point 5:   The different concrete mixtures were named as follow: How processed, unprocessed slags etc. are related with GGBFS, LFS etc.?

Response 5: Esto no sé que es lo que quiere que pongamos

¿Cómo se realizó la sustitución? ¿Por peso? ¿Han tenido los autores en cuenta el diferente peso específico de los materiales?

Point 6: How was the substitution done? By weight? Have the Authors taken into account the different specific weight of the materials?

Response 6: The cement substitution has been made by weight, this is now indicated in the materials chapter.

Point 7: Section 3: flexure and compressive=flexural and compressive strength

Response 7: The observation has been modified

Point 8: two cubic specimen of 10 cm. I know that the specimens are small, but were they compacted or vibrated?

Response 8: A phrase has been added where it is specified that the mixtures have been vibrated.

“These specimens were made according to the normative EN 12390-3 [25] and EN 12390-4 [26], the fresh mixes have been vibrated on vibrating table and they were cured in a water bath 20 °C ± 2 °C”

Point 9: centered load. It would be better a 4 points bending test, since with 3PBT the specimen breaks into the middle, while with 4PBT it breaks in the central third, where the resistance is lower

Response 9: We have used a machine that we have in the laboratory and that also meets the specifications of the regulations. For future trials we will consider your suggestion and see if we can adapt it to see if the results change.

Point 10:

chose=chosen to filter the sample. Please check the sentence better porosity. At this moment it is not clear if higher porosity means better performance so the term "better" cannot be used here. better=higher

Response 10: The observation has been modified

  Point 11:  other two electric arc slag (LFS) . Please pay attention that EAF slag is something else.and usually gives very good properties to the mixtures

Response 11: The faults detected in the slag denomination have been modified.

  Point 12:  flexion=flexure

Response 12: The observation has been modified

Point 13:  Table 5. Chromium is Cr6+? or total?

Response 13: The Chromium is total already specified in the table.

Point 14: Table 5. Clarify what CFR is.

Response 14: Before CFR you specify what it is. “the Code of Federal Regulations (CFR) 40CFR / 261.24. “

Reviewer 4 Report

The article has a very interesting topic and is well organized. The authors have provided a very concise report of their research, and the results are backed up by a wide scientific background. The emergence of the environmental risks, induced by the industry and especially from the construction industry, deserves the utmost interest by the researchers, industry and the practitioners. By this manuscript, all them may found reliable and sound justification to the utilization of the slag wastes as a practical replacer of the cement both for structural and nonstructural elements, as described in the article.

At the present form, the manuscript is publishable, once taken a careful review of the English, to free it from any grammatical errors, or typos. The main aim of the manuscript fulfill the scope of the journal and may have the potential to be a relevant contribution to the scientific community.

Author Response

Thank you very much for your contribution to the improvement of the paper, the errors detected in the text have been changed. A complete language revision has been made.

Round 2

Reviewer 1 Report

In the abstract some sentences are not complete. English check is required. Section 4.1, despite using different slag, why the densities are almost same? Section 4.2, the author linked porosity to durability which is well known fact but never explain the porosity variation. The values are simply reported without detailed investigation. Strength variation also not explained, and all the results were linked to the past studies, no novel contribution. No detailed study on the phases present in different slags and perhaps those reactions can be linked to mechanical properties. The paper feels like a testing report without much contribution after the revision.

Author Response

Point 1: In the abstract some sentences are not complete. English check is required.

Response 1:  English has been checked again

Point 2: Section 4.1, despite using different slag, why the densities are almost same? The values are simply reported without detailed investigation.

Response 2:  This point has been expanded with more details about the data obtained

Point 3: Section 4.2, the author linked porosity to durability which is well known fact but never explain the porosity variation.

Response 3:  This point has been expanded with more details about the data obtained

Point 4: Strength variation also not explained, and all the results were linked to the past studies, no novel contribution. No detailed study on the phases present in different slags and perhaps those reactions can be linked to mechanical properties. The paper feels like a testing report without much contribution after the revision. 

Response 4: As in the previous point, this information has been extended. The interesting thing in this investigation in addition to the high percentages that follow the trend that previous research is especially the comparison between the two types of streghts (compression and flexion).

Reviewer 2 Report

Some of my remarks were not taken into account, but I do not treat them as obligatory. 

Author Response

An attempt has been made to answer all your questions if there is one in particular that you want us to reinforce. We would like you to specify it. Thank you very much for your support.